# ELECTRIC CURRENTS
# FOR DISCRETE DATA GENERATION

## ABSTRACT

We propose **E**lectric **C**urrent **D**iscrete **D**ata **G**eneration (ECD$^2$G), a pioneering method for data generation in discrete settings that is grounded in electrical engineering theory. Our approach draws an analogy between electric current flow in a circuit and the transfer of probability mass between data distributions. We interpret samples from the source distribution as current input nodes of a circuit and samples from the target distribution as current output nodes. A neural network is then used to learn the electric currents to represent the probability flow in the circuit. To map the source distribution to the target, we sample from the source and transport these samples along the circuit pathways according to the learned currents. This process provably guarantees transfer between data distributions. We present proof-of-concept experiments to illustrate our ECD$^2$G method.

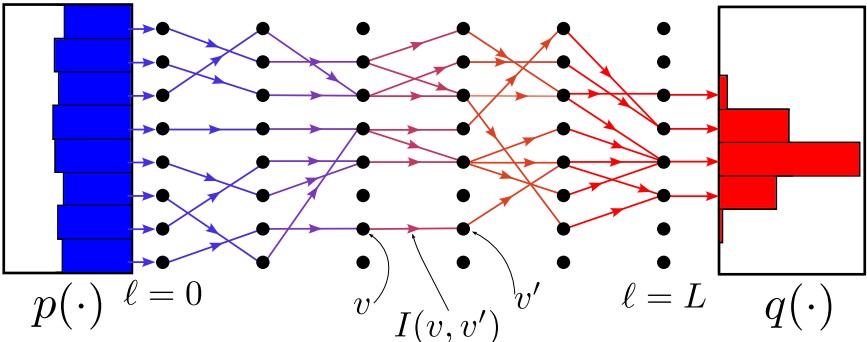

Figure 1: Our **E**lectric **C**urrent **D**iscrete **D**ata **G**eneration (ECD$^2$G) method places two data distributions $p(x_0)$ and $q(x_L)$, on the input layer $\ell = 0$ and output layer $\ell = L$ of a graph (circuit), respectively. The distribution $p(x_0)$ is interpreted as a current flowing into the circuit, and $q(x_L)$ as a current flowing out. The flow of electric current in the circuit is associated with the transfer of probability mass. As it flows through the circuit, the electric current transforms samples from the source distribution $p$ into samples from the target distribution $q$.

## 1 INTRODUCTION

Generative modeling is a challenging problem with most powerful solutions often getting a direct inspiration from concepts in **physics**. The complexity of generation is especially hard for **discrete data** (text, graphs, molecules, etc.), where the structure of the state space presents unique challenges.

Recently, **Diffusion Models** (DM) have been successfully adapted to discrete domains. While their initial remarkable success was in continuous data such as images (Sohl-Dickstein et al., 2015; Ho et al., 2020; Song & Kingma, 2021; Dhariwal & Nichol, 2021), a new wave of research has developed discrete diffusion processes (Austin et al., 2021; Lou et al., 2024; Sahoo et al., 2024). Instead of corrupting data with Gaussian noise, these models define a forward process that progressively corrupts discrete structures (e.g., tokens) through random transitions to other discrete states or a

masking token. Learning to reverse this process has led to strong performance in tasks ranging from text generation (Lou et al., 2024) to molecule design (Zhang & Zhang, 2025).

In parallel, **Electrostatic-based generative models** have emerged as a powerful physics-inspired framework for continuous data. These methods interpret data samples as electrostatic charges within an electrostatic field. Pioneering work (Xu et al., 2022; 2023) learned the field generated by data points on a plate inside a hemisphere, enabling data generation from the uniform distribution on the hemisphere. The further development (Kolesov et al., 2025; Manukhov et al., 2025) introduced a capacitor-based paradigm, learning the field between charged plates to enable data-to-data mapping.

However, a significant **gap remains**: despite their strong performance in continuous settings, electrostatic theory-based methods have not yet been formulated for generative modeling in discrete data spaces. The fundamental concepts of continuous charge and potential do not have direct analogues in discrete domains, presenting a novel and open research challenge.

**Contributions.** We introduce **E**lectric **C**urrent **D**iscrete **D**ata **G**eneration (ECD$^2$G), the first generative modeling framework for discrete data grounded in the principles of electrical engineering. We establish a novel theoretical connection between discrete data generation and current flow in a circuit, providing a new paradigm for both noise-to-data and data-to-data generation.

## 2 BACKGROUND

Let us recall the basic principles of electrical engineering to better understand the physical motivation behind our method. In this section, we introduce the graph representation of a circuit, Kirchhoff's and Ohm's rules, and the superposition principle. Further details on these electrical engineering concepts can be found in any textbook on the subject, for instance (Chen (2004), §3).

### 2.1 ELECTRIC CIRCUITS, KIRCHHOFF AND OHM LAWS.

An electric circuit can be considered as a graph $\mathcal{G}(V, E)$, where $V$ is the set of vertices (nodes) and $E$ is the set of edges, each associated with a resistor between these nodes. An edge $e(v, v') \in E$ between vertices $v \in V$ and $v' \in V$ has a resistance $R(v, v')$. We manually select a subset of vertices called *sources* and denote them by $V^{(src)} \subset V$. A current is injected into the circuit through these source vertices. We denote the current flowing into each vertex $v$ by the function $I^{(src)}(v)$, which equals zero if $v \notin V^{(src)}$.

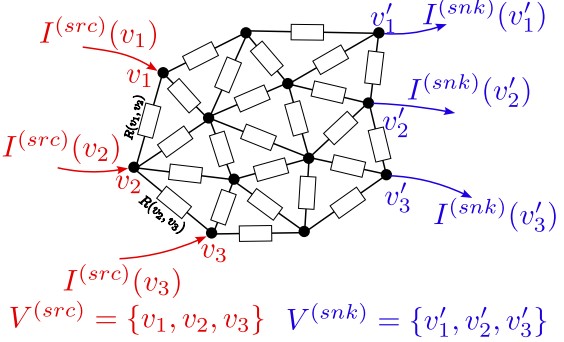

$$V^{(src)} = \{v_1, v_2, v_3\} \quad V^{(snk)} = \{v_1', v_2', v_3'\}$$

Figure 2: An illustration of an electric circuit

Similarly, we define another subset of vertices $V^{(snk)} \subset V$ called *sinks*, such that $V^{(src)} \cap V^{(snk)} = \emptyset$. We denote the currents flowing out of these sink vertices by the function $I^{(snk)}(v)$, which equals zero if $v \notin V^{(snk)}$. As current flows from the sources $V^{(src)}$ to $V^{(snk)}$, they obey the following physical laws.

**Kirchhoff's Current Law.** Consider any vertex $v \in V$. There exists a set of edges $\{e(v, v')\}_{v' \in J(v)}$ connecting $v$ to all its neighboring vertices, where the set of these neighbors is denoted by $J(v)$. Let $J_{in}(v)$ be the set of vertices from which current flows into $v$, and let $J_{out}(v)$ be the set of vertices to which current flows out from $v$.

If the current between $v$ and a vertex $v' \in J(v)$ is equal to zero, we assign $v'$ to $J_{in}(v)$ by default. The sets $J_{in}(v)$ and $J_{out}(v)$ are disjoint and their union is $J(v) = J_{in}(v) \cup J_{out}(v)$.

The Kirchhoff's law states that the algebraic sum of all currents entering any node must equal zero. Equivalently, the total current entering a node equals the total current leaving it. For any node $v \notin V^{(src)} \cup V^{(snk)}$:

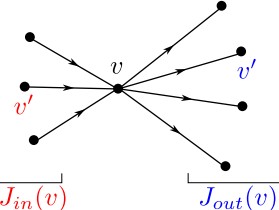

Figure 3: An illustration for Kirchhoff's Law

$$\sum_{v' \in J_{in}(v)} I(v', v) = \sum_{v' \in J_{out}(v)} I(v, v'), \qquad (1)$$

where $I(v, v')$ denotes the current flowing from $v$ to $v'$. If a node $v \in V^{(src)}$ or $v \in V^{(snk)}$ is considered, then equation (1) becomes:

$$I^{(src)}(v) = \sum_{v' \in J_{out}(v)} I(v, v'), \quad I^{(snk)}(v) = \sum_{v' \in J_{in}(v)} I(v', v). \qquad (2)$$

A direct consequence of applying the Kirchhoff's rule to every node in the circuit is that the total current entering the entire circuit must equal the total current leaving it. For our circuit with a source vertex set $V^{(src)}$ and a sink vertex set $V^{(snk)}$,

$$\sum_{v \in V^{(src)}} I^{(src)}(v) = \sum_{v \in V^{(snk)}} I^{(snk)}(v), \qquad (3)$$

where the left side of the equation represents the total injected current at the source nodes while the r.h.s. is the total extracted current at the sink nodes.

**Ohm's law.** In a circuit, there exists a function $\phi : V \to \mathbb{R}$ called electric potential, defined up to a constant that fully describes the set of currents in the circuit. For arbitrary neighboring nodes $v$ and $v'$ in the circuit $\phi$ satisfy:

$$\frac{\phi(v) - \phi(v')}{R(v, v')} = I(v, v'), \qquad (4)$$

where $R(v, v')$ and $I(v, v')$ are the resistance and the current between the nodes, respectively[1].

**Superposition principle.** Consider two systems of sources and sinks on $\mathcal{G}(V, E)$: $\{I_1^{(src)}(v)\}_{v \in V_1^{(src)}}$, $\{I_1^{(snk)}(v)\}_{v \in V_1^{(snk)}}$ and $\{I_2^{(src)}(v)\}_{v \in V_2^{(src)}}$, $\{I_2^{(snk)}(v)\}_{v \in V_2^{(snk)}}$. Suppose the first system establishes potentials $\phi_1(v), v \in V$, and the second system establishes potentials $\phi_2(v)$. Then, the two systems merged together will produce a potential

$$\phi(v) = \phi_1(v) + \phi_2(v), \qquad (5)$$

which also means that current in the merged system is additive for all $v, v' \in V$:

$$I(v, v') = I_1(v, v') + I_2(v, v'). \qquad (6)$$

## 3 ELECTRIC CURRENTS DISCRETE DATA GENERATION (ECD²G)

This section introduces Electric Currents Discrete Data Generation (ECD²G), a novel generative modeling paradigm grounded in electric engineering for discrete data settings. In §3.1, we formulate the task of distribution transfer and give an intuitive description of our proposed method. In §3.2, we provide the theoretical foundation of ECD²G and the main propositions for a connected graph. In Sections §3.3 and §3.4, we give specific graph structure for an electric circuit and formulate the learning and inference algorithms for this graph, respectively.

---

[1]The difference in potentials $\phi(v) - \phi(v')$ for neighboring vertices is called the *voltage* and denoted as $U(v, v')$. Eq. (4) can be rewritten as $\frac{U(v, v')}{R(v, v')} = I(v, v')$.

## 3.1 PROBLEM FORMULATION AND AN INTUITIVE EXPLANATION OF OUR ECD$^2$G METHOD

**Problem formulation.** We assume that the state space $\mathcal{X}$ is a discrete space $\mathcal{X} = \mathbb{S}^D$, where $\mathbb{S}$ is a finite set. For convenience, we say that it is the space of categories, e.g., $\mathbb{S} = \{1, 2, .., S\}$. We denote a set of probability distributions on this space as $\mathcal{P}(\mathbb{S}^D)$ and introduce a *source* distribution as $p \in \mathcal{P}(\mathbb{S}^D)$ and a *target* distribution as $q \in \mathcal{P}(\mathbb{S}^D)$. These data distributions are assumed to be accessible by random empirical samples $x_1, ..., x_N \sim p$ and $y_1, ..., y_M \sim q$ that are $D$-dimensional vectors of categories. **The task of distribution transfer** is to use these empirical samples to build a procedure $T$ that is capable to transform any new samples $x^{new} \sim p$ in to samples from $q$. Formally, this is equivalent to samples $T(x^{new})$ being distributed as the target samples from $q$.

**Intuitive explanation of our ECD$^2$G approach.** We construct a current circuit $\mathcal{G}(V, E)$ with $V^{(src)} = \mathcal{X}$ and $V^{(snk)} = \mathcal{X}'$, where $\mathcal{X}'$ is an independent copy of $\mathcal{X}$. We interpret a current $I^{(src)}(v)$ flowing into a circuit at a subset of states $\mathcal{X}$ as the source distribution $p(x)$, where $x \in \mathcal{X}$. Analogously, a current $I^{(snk)}(v)$ flowing out of the circuit through states $\mathcal{X}'$ is associated with the target distribution $q(x')$, where $x' \in \mathcal{X}'$. We approximate the currents using a neural-network-based approach. By associating the electric current in the circuit with the transfer of probability mass, we map samples $x^{new} \sim p$ through the circuit to the target distribution $q$.

## 3.2 MAIN THEORETICAL CONSTRUCTION OF OUR ECD$^2$G METHOD

For convenience, we consider a general electric circuit as a connected graph $\mathcal{G}(V, E)$. However, a particular practical realization of the circuit is given in the next section (§3.3). We assume that each edge $e(v, v') \in E$ has a resistance $R(v, v')$. Recall that from Kirchhoff's law (3) the total incoming current $I^{(src)}$ equals the total outgoing current $I^{(snk)}$. Without a loss of generality, we suppose that the total incoming current as well as the total outgoing current are each equal to 1:

$$\sum_{v \in V^{(src)}} I^{(src)}(v) = \sum_{v' \in V^{(snk)}} I^{(snk)}(v') = 1. \tag{7}$$

This allows to interpret $I^{(src)}$ and $I^{(snk)}$ as the source distribution $p$ and the target $q$, respectively:

$$\forall v \in V^{(src)} : I^{(src)}(v) = p(v) \quad \text{and} \quad \forall v' \in V^{(snk)} : I^{(snk)}(v') = q(v'). \tag{8}$$

The transfer of probability mass from source to sink is realized by the stochastic walk on the circuit. We set up the following rule for the steps between neighboring vertices.

**Definition 3.1** (Movement Rule)**.** *The probability of transition from vertex $v \in V : v \notin V^{(snk)}$ to a neighbor $v' \in J_{out}(v)$ is defined by:*

$$\text{prob}(v'|v) \stackrel{def}{=} \frac{I(v, v')}{\sum\limits_{v'' \in J_{out}(v)} I(v, v'')}. \tag{9}$$

*If $v \in V^{(snk)}$, the flow of current either terminates with probability* $\text{prob}(term|v)$*, or, if possible, continues to vertex $v'$ with probability* $\text{prob}(v'|v)$ *as follows:*

$$\text{prob}(term|v) \stackrel{def}{=} \frac{I^{(snk)}(v)}{I^{(snk)}(v) + \sum\limits_{v'' \in J_{out}(v)} I(v, v'')}, \quad \text{prob}(v'|v) \stackrel{def}{=} \frac{I(v, v')}{I^{(snk)}(v) + \sum\limits_{v'' \in J_{out}(v)} I(v, v'')}. \tag{10}$$

The denominator represents the total current flowing out of vertex $v$. Therefore, the probabilities (9) and (10) of movement to neighboring vertices are proportional to the current in the respective edges. Having determined one-step transition (9) in the circuit, we note that the following termination property holds for the multi-step transition in the circuit.

**Proposition 3.2** (Finite-step termination)**.** *If the movement through the circuit starts at vertices $V^{(src)}$ and obeys the movement rule 3.1, it terminates at nodes $V^{(snk)}$ in a finite number of steps.*

*Proof.* Since the graph is finite, infinite movement is only possible if there exists a cycle of vertices $v_1, v_2, ..., v_C$, where $v_C = v_1$ with non-zero current. However, current flows from higher to lower potential. For a cycle to sustain a current, the potentials must satisfy $\phi(v_1) > \phi(v_2) > ... > \phi(v_C) = \phi_1$, which is a contradiction. Thus, the movement consists of a finite number of steps. $\quad\square$

Proposition 3.2 allows us to define a *stochastic* mapping $T : V^{(src)} \to V^{(snk)}$ that transforms sources to sinks according to movement rule 3.1. We now formulate the proposition, which guarantees a successful transfer between the source and target distributions.:

**Corollary 3.3** (ECD$^2$G transport)**.** *Considering the aforementioned circuit scheme, the sequential movement by $T$ from $v \in V^{(src)}$ terminates at $T(v) \in V^{(snk)}$ with probability 1. Moreover, if $v \sim p$ is a random variable, then $T(v) \sim q$, i.e. the movement through the circuit with probabilities proportional to the currents transforms distribution $p$ on $V^{(src)}$ into distribution $q$ on $V^{(snk)}$.*

This result follows directly from classical network flow theory (Ahuja et al., 1993, §6) and properties of absorbing Markov chains (Grinstead & Snell, 1997, §11). The conservation (1) and normalization (7) conditions establish a feasible unit flow from sources $V^{(src)}$ to sinks $V^{(snk)}$. The movement rule (9) defines an absorbing Markov chain where sinks are absorbing states. For a rigorous proof of this absorption property for Markov chains defined by unit flows see, e.g., (Kemeny & Snell, 1960, §3).

### 3.3 PRACTICAL REALIZATION OF AN ELECTRIC CIRCUIT

In practice, we are interested in transforming distributions on a particular discrete space $\mathcal{X}$. Thus, we consider a specific structure of an electric circuit as an $L$-partite graph $\mathcal{G}(V, E)$. This graph consists of $|V| = (L+1)|\mathcal{X}|$ vertices, where each partite (layer) is composed of $|\mathcal{X}|$ vertices. We denote the set of vertices on the $\ell$-th layer as $\mathcal{X}_{(\ell)}$, then $V = \mathcal{X}_{(0)} \cup .. \cup \mathcal{X}_{(L)}$. Edges $E$ of this graph can only connect vertices from neighboring layers, that is, $E = \cup_{\ell=0}^{L-1} \mathcal{X}_{(\ell)} \times \mathcal{X}_{(\ell+1)}$. In this configuration (Figure 1), the distribution $p$ is defined on $\mathcal{X}_{(0)} \equiv V^{(src)}$, while the distribution $q$ is defined on $\mathcal{X}_{(L)} \equiv V^{(snk)}$. A vertex $x_0 \in \mathcal{X}^{(0)}$ is a source vertex in $V^{(src)}$ where a current $I^{(src)}(x_0) = p(x_0)$ enters. Similarly, $x_L \in \mathcal{X}_{(L)}$ is a sink in $V^{(snk)}$ where a current $I^{(snk)}(x_L) = q(x_L)$ quits.

We denote the potentials, currents, and resistances with an additional layer index $\ell$ as $\phi_\ell(x_\ell),\ R_{\ell,\ell+1}(x_\ell, x_{\ell+1}),\ I_{\ell,\ell+1}(x_\ell, x_{\ell+1})$ . We set the resistances as follows:

$$R_{\ell,\ell+1}(x_\ell, x_{\ell+1}) = \begin{cases} r, & x_\ell = x_{\ell+1} \\ R, & x_\ell \neq x_{\ell+1} \end{cases} \tag{11}$$

In other words, the "forward" resistance is $r$, and all "lateral" resistances are equal and denoted by $R$. Thus, using $R > r$ in practice, we encourage the preservation of the state in the next layer rather than its change. By introducing the resistances $r$ and $R$ into the circuit, we can now define the electrical circuit as an $L$-partite graph $\mathcal{G}_L(|\mathcal{X}|, R, r)$.

Calculation of potentials $\phi_l$ for $\mathcal{G}_L(|\mathcal{X}|, R, r)$ and arbitrary current distributions $p,q$ is a challenging task, especially, when the distributions are only available through a limited amount of empirical samples (§3.1). However, using the superposition principle (5), it can be reduced to solving the problem for a single unit input current at $x_0 \in \mathcal{X}_{(0)}$ which flows out at $x_L \in \mathcal{X}_{(L)}$. Let $\phi_\ell(x_\ell|x_0, x_L)$ define the potential created in point $x_\ell$ in such system. Then, the superposition principle yields

$$\phi_\ell(x_\ell) = \sum_{x_0, x_L} \phi_\ell(x_\ell|x_0, x_L) \pi(x_0, x_L), \tag{12}$$

where $\pi(x_0, x_L)$ is an arbitrary joint distribution on $\mathcal{X}_{(0)} \times \mathcal{X}_{(L)}$ satisfying $\sum_{x_L} \pi(x_0, x_L) = p(x_0)$ and $\sum_{x_0} \pi(x_0, x_L) = q(x_L)$. Since the right hand side of (12), i.e., the potential $\phi_\ell(x_\ell)$, is determined (up to a constant) uniquely by the marginal input $I^{(src)}(x_0) = p(x_0)$ and output $I^{(snk)}(x_L) = q(x_L)$ current distributions, the following observation arises.

**Remark 3.4** (Transport plan independence)**.** *Potential $\phi_\ell(x_\ell)$ is independent from $\pi(x_0, x_L)$.*

Our following results provides the formula for potential $\phi(x_l|x_0, x_L)$ in one source/one sink system.

**Lemma 3.5** (Single source-sink potentials)**.** *Let a unit current flow into the electrical circuit $\mathcal{G}_L(|\mathcal{X}|, R, r)$ at position $x_0 \in \mathcal{X}^{(0)}$ and flow out at position $x_L \in \mathcal{X}^{(L)}$. Then the potentials $\phi_\ell(x_\ell|x_0, x_1)$ at the vertices for each layer $\ell$ of this graph are given in Table 1.*

The proof of the Lemma is given in Appendix A. Finally, using the Ohm's law (4), Lemma 3.5 and the superposition principle (5), (6), one may compute the electric current between the $x_\ell, x_{\ell+1}$:

$$I_{\ell,\ell+1}(x_\ell, x_{\ell+1}) = \frac{\phi_\ell(x_\ell) - \phi_{\ell+1}(x_{\ell+1})}{R_{\ell,\ell+1}(x_\ell, x_{\ell+1})} = \mathbb{E}_{(x_0, x_L) \sim \pi}\left[ \frac{\phi_\ell(x_\ell|x_0, x_L) - \phi_{\ell+1}(x_{\ell+1}|x_0, x_L)}{R_{\ell,\ell+1}(x_\ell, x_{\ell+1})} \right]. \tag{13}$$

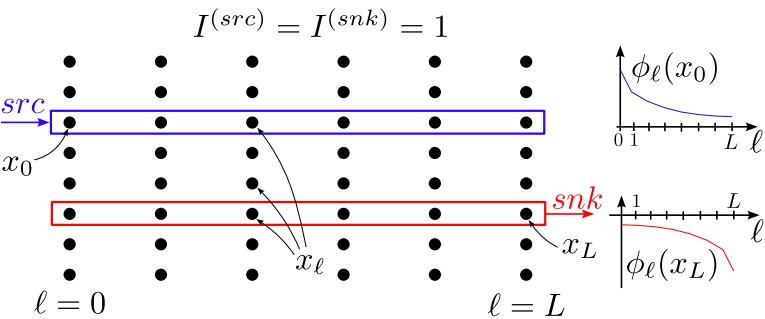

Figure 4: A sketch of single source-sink configuration for electric circuit $\mathcal{G}_L(|\mathcal{X}|, R, r)$.

| **Case** $x_0 \neq x_L$ | **Case** $x_0 = x_L$ |
|---|---|
| $\phi_\ell(x_\ell) = \begin{cases} \frac{T_{L-\ell}(\gamma)}{\gamma T_L(\gamma) - T_{L-1}(\gamma)} \cdot r & \text{if } x_\ell = x_0 \\ -\frac{T_\ell(\gamma)}{\gamma T_L(\gamma) - T_{L-1}(\gamma)} \cdot r & \text{if } x_\ell = x_L \\ 0 & \text{otherwise} \end{cases}$ | $\phi_\ell(x_\ell) = \begin{cases} \left[ T_\ell(\gamma) \frac{\gamma U_{L-1}(\gamma) - U_{L-2}(\gamma) - 1}{\gamma T_L(\gamma) - T_{L-1}(\gamma)} - \right. \\ \left. - U_{\ell-1}(\gamma) \right] r & \text{if } x_\ell = x_0 \equiv x_L \\ 0 & \text{otherwise} \end{cases}$ |

Table 1: Single source-sink potentials. Here $\gamma = 1 + \frac{r}{R}|\mathcal{X}|$, and $T_l(\cdot)$, $U_l(\cdot)$ are Chebyshev polynomials of the 1st and 2nd kind, respectively. The parameter $\gamma$ controls the rate at which the potential decreases across layers. If $\gamma \to 1+0$, the states remain almost unchanged between layers; otherwise when $\gamma \gg 1$, the change occurs faster.

Thanks to our derived closed form for $\phi_\ell(x_\ell | x_0, x_L)$, we can now estimate the current using random samples $(x_0, x_L) \sim \pi$ for any coupling $\pi$ of $p$ and $q$. This is the key ingredient of our method.

### 3.4 LEARNING AND INFERENCE ALGORITHMS

**Training** (Algorithm 1). Recall that $\mathcal{X} = \mathbb{S}^D$ in practice. To recover currents $I_{\ell,\ell+1}(x_\ell, x_{\ell+1})$ in a circuit, we approximate them using a neural net $I_\theta(\cdot) : \mathbb{S}^D \times \mathbb{S}^D \times \overline{0, L-1} \to \mathbb{R}$. It takes as input the states (nodes in the circuit) $x_\ell \in \mathcal{X}_{(\ell)}$ and $x_{\ell+1} \in \mathcal{X}_{(\ell+1)}$, and the layer number $\ell \in \overline{0, L-1}$ of the current state as a condition. The output of the network is a value $I_\theta(x_\ell, x_{\ell+1}, \ell) \approx I_{\ell,\ell+1}(x_\ell, x_{\ell+1})$.

Since the circuit consists of $LS^D$ nodes, considering all possible pairs of nodes in neighboring layers for current recovery is an infeasible task. Therefore, we randomly sample them for training. Namely, we sample the layer $\ell$ from the discrete uniform distribution on $\overline{0, L-1}$ and randomly select states $x_\ell \in \mathcal{X}_{(\ell)}$ and $x_{\ell+1} \in \mathcal{X}_{(\ell+1)}$ from $\mathcal{U}(S^D)$. The ground-truth current $I_{\ell,\ell+1}(x_\ell, x_{\ell+1})$ is estimated using (13). We additionally sample a random batch $(x_0, x_L) \sim p \times q$ to obtain a Monte-Carlo estimate $\widehat{I}_{\ell,\ell+1}(x_\ell, x_{\ell+1})$ for (13). Then we regress a neural network $I_\theta$ to approximate the current. The resulting loss is given by:

---

**Algorithm 1** ECD²G Training

**Input:** Distributions $p(x)$ and $q(y)$ accessible by samples;
    Resistances $r > 0$ and $R > 0$ ;
    $L$-partite graph (circuit) $\mathcal{G}_L(|\mathbb{S}^D|, R, r)$ ;
    NN approximator $I_\theta$.
**Output:** The learned currents $I_\theta$
**Repeat until converged:**
    Sample batches of $x \sim p$ and $y \sim q$;
    Set $x_0 \equiv x$ and $x_L \equiv y$ on $\mathcal{G}_L(|\mathbb{S}^D|, R, r)$
    Sample batch of layer numbers $\ell \sim \mathcal{U}(L-1)$ ;
    Sample batch of states $x_\ell, x_{\ell+1} \sim \mathcal{U}(S^D)$ ;
    Approximate $I_{\ell,\ell+1}(x_\ell, x_{\ell+1})$ with $(x_0, x_L)$ using (13);
    Minimize MSE b/w $I_\theta(x_\ell, x_{\ell+1}, l)$ and approx. current;

---

$$\mathbb{E}_{\ell \sim \mathcal{U}(L-1)} \mathbb{E}_{x_\ell, x_{\ell+1} \sim \mathcal{U}(S^D)} ||I_\theta(x_\ell, x_{\ell+1}, \ell) - \widehat{I}_{\ell,\ell+1}(x_\ell, x_{\ell+1})||_2^2 \to \min_\theta. \quad (14)$$

**Inference.** For modeling the current flow starting at any input $x = x_0 \sim p$, we simply use the movement rule from Definition 3.1, where we compute all the terms explicitly. For simplicity, we

model only forward current direction, i.e., the layer number $\ell$ only grows. Also we always terminate the movement at the last layer $\ell = L$.

# 4 RELATED WORKS

**Generative Flow Networks** (Malkin et al., 2022, GFlowNets) paradigm is a family of models that learn to *sample* discrete structures with a probability proportional to a given reward function. Although, the problem they consider differs from ours, their approach shares similarities with our work. They also look at the flow of probability mass through a graph. However, they learn edges of a graph unlike our method, where we first predefine resistance of edges and then compute the flow.

**Discrete Diffusion Models** (Hoogeboom et al., 2021; Lou et al., 2024; Gat et al., 2024, DDM) are powerful classes for data generation in discrete data setups. DDM use forward (data destruction) process built on *predefined* transition matrix unlike our method. As a result, the forward process of DDM via uniform (Austin et al., 2021) or absorbing (Sahoo et al., 2024) transition matrix noises clean data either into a uniformly distributed sample or a special token. In contrast, our approach is theoretically able to learn transformations between any discrete data distributions.

**Discrete Flow Matching** (Gat et al., 2024, DFM) learns a flow field between two data distributions. In contrast to our ECD$^2$G, their flow learned in their DFM depends on the data transport plan $\pi$ used for training, while our flow is the uniquely determined flow of the electric current (Remark 3.4). Note, in our method, one may estimate the current in any edge $(x_\ell, x_{\ell+1})$ with arbitrary batch size of samples from $\pi$. In turn, in DFM, and, generally, FM, only one-sample estimates are available, and only for points specifically sampled from the interpolant (Ryzhakov et al., 2024, §1.2).

**Electrostatic-based methods** (Xu et al., 2023; Kolesov et al., 2025; Manukhov et al., 2025) interpret data distributions $p(\cdot)$ and $q(\cdot)$ as positive and negative charge densities, respectively. Data transport and/or generation are performed by moving along the field lines generated by these charge distributions. The theoretical justification for these methods relies on the principle of flux conservation in the generated field. The proposed ECD$^2$G model also employs electricity theory, where flux conservation corresponds to the Current Law (1). However, all existing electrostatic-based models operate in a continuous setting, whereas ECD$^2$G performs discrete data generation.

# 5 EXPERIMENTAL ILLUSTRATIONS

In this section, we present proof-of-concept experiments for our proposed method, ECD$^2$G. We demonstrate a 1-dimensional transfer task in §5.1 as well as a 2-dimensional one in §5.2. Further experimental details are provided in Appendix B.

## 5.1 1D EXPERIMENT

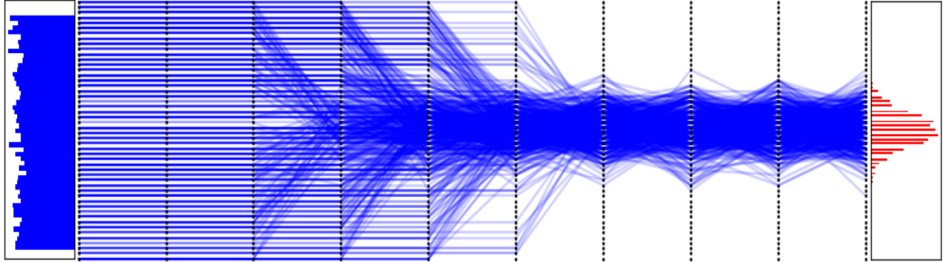

Figure 5: *Illustrative 1D Uniform→Gaussian experiment*: input and target samples $x_0 \sim p(x_0)$ and $x_{10} \sim q(x_{10})$, along with the **ground-truth** currents obtained from (13) on $\mathcal{G}_{10}(50, 100, 0.1)$.

The first intuitive test to validate our approach is to solve a transfer task between two 1-dimensional data distributions, whose densities and current flows can be visualized. We consider a one-dimensional uniform distribution $\mathcal{U}(0, 50)$ as the *source* $p$ and a Gaussian distribution $\mathcal{N}(25, 1)$ as the *target* $q$. Both distributions are discretized into $S = 50$ categories; their visualizations are shown in Fig. 5. We represent the circuit between $p$ and $q$ as an $L$-partite graph $\mathcal{G}_L(50, R, r)$ with $L = 10$ layers, where resistances $r$ and $R$ are 0.1 and 100, respectively.

We train a neural network $I_\theta$ with our algorithm 1 and evaluate it by the inference (see §3.4) stage. Our approach maps to the target density quite well, recovering the target distribution (see Fig. 6). We also compare the flow of the ground-truth currents by (13) with the currents learned by $I_\theta$. The learned current flow demonstrates similar behavior but with a greater stochasticity.

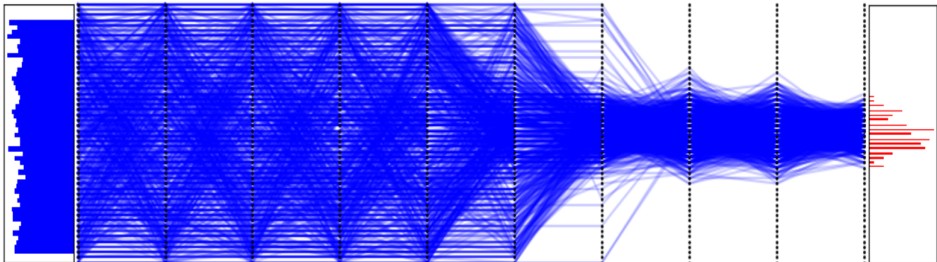

Figure 6: *Illustrative 1D Uniform→Gaussian experiment*: input and their mapped by $T(\cdot)$ samples $x_0 \sim p(x_0)$ and $x_L \sim T(x_0)$, along with the currents **learned** by $I_\theta$ on $\mathcal{G}_{10}(50, 100, 0.1)$.

### 5.2 2D EXPERIMENT: MOONS TO SWISS ROLL EXPERIMENT

We also provide a proof-of-concept for our method by transferring between two 2-dimensional data distributions. We consider the 2-dimensional Moons dataset as the source distribution $p$ and the Swiss Roll dataset as the target $q$; their visualizations are shown in Fig. 7. Both datasets are discretized into $S = 50$ categories, resulting in a 2-dimensional categorical space of $S^D = 50^2 = 2500$ points. The graph structure for this experiment is $\mathcal{G}_L(2500, R, r)$ with $L = 4$, $R = 10$ and $r = 0.1$. Our approach maps to the target density effectively, successfully recovering the target distribution as shown in Fig. 7.

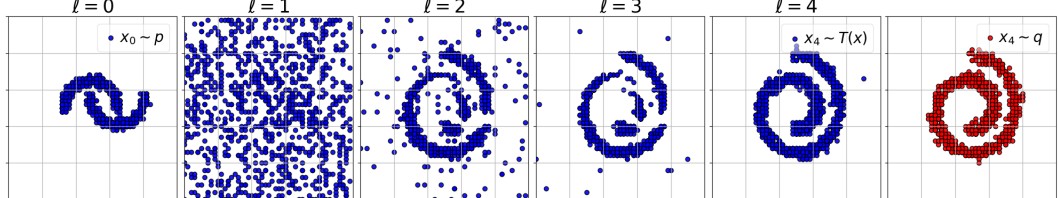

Figure 7: *Illustrative 2D Moons→Swiss Roll experiment*: input and target distributions $p(x_0)$ and $q(x_4)$, together with the intermediate distributions at layers $\ell = \overline{1, 3}$ and samples transformed from $x \sim p$ by $T(x)$ in a circuit $\mathcal{G}_4(2500, 10, 0.1)$.

For completeness, we provide quantitative results for our method on the 2D experiment: *Moons→Swiss roll*, comparing its performance with the baselines (Austin et al., 2021, **D3PM**) and (Gat et al., 2024, **DFM**). We use the Maximum Mean Discrepancy (MMD) metric with a Gaussian kernel (Gretton et al., 2012) between 2048 generated and original samples from the *Swiss roll* for a quantitative comparison. We report mean and standard deviation of MMD for the corresponding methods over 5 runs in Table 2.

|  | ECD$^2$G | D3PM | DFM |
|---|---|---|---|
| MMD(x 1e-3) | 1.24 ± 0.18 | 1.67 ± 0.15 | 2.32 ± 0.22 |

Table 2: The mean and standard deviation of the MMD metric for the ECD$^2$G, D3PM, and DFM approaches, calculated over 5 runs with a sample size 2048.

## 6 FUTURE WORK

**Defining the optimal training volume.** Our training approach involves sampling layers $\ell$ and states $x_\ell$ within these layers. We use uniform discrete distributions over $L$ layers and $S^D$ categories (see §3.4). This allows us to approximate the current in the circuit between a state $x_\ell$ and states $x_{\ell+1}$.

However, this sampling is not unique, and there are many other possible ways to define a training volume for efficient learning. This is a promising direction for further research.

**Circuit construction.** Circuit scheme and edge resistances are the key hyperparameters in our approach. Our proposed current flow model and its resistance assignment scheme encourage flow to remain at the same state in the next layer, while slightly suppressing flow to other nodes. Future work could explore resistance schemes that assign distinct resistances to each node pair. More broadly, developing more efficient circuit architectures remains a promising research direction.

**Factorization.** We perform low-dimensional experiments where we can explicitly perform each sampling step using (9). However, the denominator in (9) likely becomes computationally infeasible as the dimensionality and number of categories increase. Following discrete diffusions (Lou et al., 2024; Campbell et al., 2022), one could explore a factorization technique to improve the method's scalability, which represents an interesting direction for future work.

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

# A  SINGLE SOURCE-SINK SETTING

## A.1  CONDUCTANCE MATRIX AND THE NODE-VOLTAGE EQUATION IN A MATRIX FORM

To solve the formulated problem, we need to define the conductance matrix and write down the node-voltage equation in a matrix form.

**Definition A.1** (Conductance Matrix). *The conductance matrix $G$, defined for an electrical circuit represented by a graph $G(V, E)$, is a matrix of size $|V| \times |V|$, where:*

$$
\begin{aligned}
G_{ii} &= \sum_{v_k \in J_n(v_i)} \frac{1}{R(v_i, v_k)} \quad \text{for diagonal terms,} \\
G_{ik} &= -\frac{1}{R(v_i, v_k)} \quad \text{for non-diagonal terms } (i \neq k).
\end{aligned}
\tag{15}
$$

Let us define the vector containing all node potentials:

$$
\phi = \begin{pmatrix} \phi(v_1) \\ \phi(v_2) \\ \vdots \\ \phi(v_{|V|}) \end{pmatrix}.
\tag{16}
$$

And the vector of currents:

$$
i = \begin{pmatrix} i(v_1) \\ i(v_2) \\ \vdots \\ i(v_{|V|}) \end{pmatrix},
\tag{17}
$$

where $i(v_k) = \sum_{v \in J_n(v_k)} I(v, v_k)$. If a current $I^{(\text{src})}(v_k)$ flows *into* node $v_k$, then $i(v_k) = I^{(\text{src})}(v_k)$; if a current $I^{(\text{snk})}(v_k)$ flows *out of* node $v_k$, then $i(v_k) = -I^{(\text{snk})}(v_k)$. In all other cases, by Kirchhoff's Current Law, $i(v_k) = 0$, since the sum of currents flowing into a node equals the sum of currents flowing out.

Thus, the successive application of Ohm's and Kirchhoff's laws leads to the equation for determining the node potentials, known as the **node-voltage equation**:

$$
G \cdot \phi = i.
\tag{18}
$$

## A.2  ANALYTICAL SOLUTION FOR $\mathcal{G}_L(|\mathcal{X}|, R, r)$

Consider an electrical circuit $\mathcal{G}_L(|\mathcal{X}|, R, r)$ with a unit current entering the system at $x_0 \in \mathcal{X}_{(0)}$ and leaving at $x_L \in \mathcal{X}_{(L)}$. Assume there are no other sources or sinks in the circuit besides $x_0$ and $x_L$. Each layer $\mathcal{X}_\ell$ of the graph $\mathcal{G}_L(|\mathcal{X}|, R, r)$ is a copy of $\mathcal{X} = \mathbb{S}^D$. Consequently, two distinct situations may arise: $x_0 = x_L$ or $x_0 \neq x_L$. These cases must be analyzed separately.

### A.2.1  STRUCTURE OF THE POTENTIAL $\phi$ AND CURRENT $i$ VECTORS

The graph $\mathcal{G}_L(|\mathcal{X}|, R, r)$ consists of $L + 1$ layers. Each layer contains $n = |\mathcal{X}|$ vertices. Therefore, we will express the potential vector $\phi$ and the current vector $i$ in a block structure:

$$
\phi = \begin{pmatrix} \phi_0 \\ \phi_1 \\ \vdots \\ \phi_L \end{pmatrix}, \quad i = \begin{pmatrix} i_0 \\ i_1 \\ \vdots \\ i_L \end{pmatrix}.
\tag{19}
$$

Here, each $\phi_\ell$ and $i_\ell$ is a column vector of height $n$:

$$\forall \ell = 0, \ldots, L: \quad \phi_\ell = \begin{pmatrix} \phi_\ell(x_{\ell 1}) \\ \phi_\ell(x_{\ell 2}) \\ \vdots \\ \phi_\ell(x_{\ell n}) \end{pmatrix}, \quad i_\ell = \begin{pmatrix} i_\ell(x_{\ell 1}) \\ i_\ell(x_{\ell 2}) \\ \vdots \\ i_\ell(x_{\ell n}) \end{pmatrix}. \tag{20}$$

### A.2.2 SYMMETRY OF THE PROBLEM

The problem possesses a high degree of symmetry. In formulas (19) and (20), we considered "vertical" cross-sections of the graph, i.e., layers. Let us now consider a "horizontal" cross-section, namely, the sets of values:

$$(\phi_0(x'), \ \phi_1(x'), \ \phi_2(x'), \ \ldots, \ \phi_L(x')), \quad \text{where } x' \in \mathcal{X} \text{ is fixed.} \tag{21}$$

These sets are entirely equivalent for all $x' \neq x_0$ and $x' \neq x_L$. Therefore, the problem of finding $n(L+1)$ unknown potentials reduces to finding only $3(L+1)$ potentials at the points:

$$\begin{cases} \phi_\ell(x_0), & \ell = 0, \ldots, L; \\ \phi_\ell(x_L), & \ell = 0, \ldots, L; \\ \phi_\ell(x'), & \ell = 0, \ldots, L, \text{ where } x' \neq x_0, x_L. \end{cases} \tag{22}$$

### A.2.3 STRUCTURE OF THE CONDUCTANCE MATRIX AND THE NODE-VOLTAGE EQUATION

For the resistances defined in (11), the conductance matrix takes the form of a block-tridiagonal matrix:

$$G = \begin{pmatrix} D & B & 0 & 0 & \cdots & 0 \\ B^T & 2D & B & 0 & \cdots & 0 \\ 0 & B^T & 2D & B & \cdots & 0 \\ 0 & 0 & B^T & 2D & \ddots & \vdots \\ \vdots & \vdots & \vdots & \ddots & \ddots & B \\ 0 & 0 & 0 & \cdots & B^T & D \end{pmatrix}. \tag{23}$$

Here, $B$ and $D$ are $n \times n$ matrices, $n = |\mathcal{X}|$, with the following form:

$$D = \left( \frac{1}{r} + \frac{n-1}{R} \right) \cdot \begin{pmatrix} 1 & 0 & .. & 0 \\ 0 & 1 & .. & 0 \\ .. & .. & .. & .. \\ 0 & 0 & .. & 1 \end{pmatrix}; \quad B = \begin{pmatrix} -\frac{1}{r} & -\frac{1}{R} & -\frac{1}{R} & \cdots & -\frac{1}{R} \\ -\frac{1}{R} & -\frac{1}{r} & -\frac{1}{R} & \cdots & -\frac{1}{R} \\ -\frac{1}{R} & -\frac{1}{R} & -\frac{1}{r} & \cdots & -\frac{1}{R} \\ \vdots & \vdots & \vdots & \ddots & \vdots \\ -\frac{1}{R} & -\frac{1}{R} & -\frac{1}{R} & \cdots & -\frac{1}{r} \end{pmatrix}. \tag{24}$$

Given that all current vectors $i_1, i_2, \ldots, i_{L-1}$ are identically equal to zero (since the sum of incoming and outgoing currents is balanced at the vertices of the intermediate layers), equation (18) takes the following form:

$$\begin{cases} D\phi_0 + B\phi_1 & = i_0, \\ B^T\phi_{\ell-1} + 2D\phi_\ell + B\phi_{\ell+1} & = 0, \quad \forall \ell = 1, \ldots, L-1, \\ B^T\phi_{L-1} + D\phi_L & = i_L. \end{cases} \tag{25}$$

### A.2.4 Case $x_0 \neq x_L$

Now we are finally ready to consider the cases of coinciding and non-coinciding source and sink.

Equation (25) in this case takes the form of the following system of equations:

The first set (corresponding to $D\phi_0 + B\phi_1 = i_0$):

$$
\begin{cases}
\left(\frac{1}{r} + \frac{n-1}{R}\right)\phi_0(x_0) - \frac{1}{r}\phi_1(x_0) - \frac{n-2}{R}\phi_1(x') - \frac{1}{R}\phi_1(x_L) = 1, \\
\left(\frac{1}{r} + \frac{n-1}{R}\right)\phi_0(x') - \frac{1}{r}\phi_1(x_0) - \left(\frac{1}{r} + \frac{n-3}{R}\right)\phi_1(x') - \frac{1}{R}\phi_1(x_L) = 0, \\
\left(\frac{1}{r} + \frac{n-1}{R}\right)\phi_0(x_L) - \frac{1}{r}\phi_1(x_L) - \frac{n-2}{R}\phi_1(x') - \frac{1}{R}\phi_1(x_0) = 0.
\end{cases}
\tag{26}
$$

Next, for all $\ell = 1, 2, \ldots, L-1$ (corresponding to $B^T\phi_{\ell-1} + 2D\phi_\ell + B\phi_{\ell+1} = 0$):

$$
\begin{cases}
-\frac{1}{r}\phi_{\ell-1}(x_0) - \frac{1}{R}\phi_{\ell-1}(x_L) - \frac{n-2}{R}\phi_{\ell-1}(x') \\
\quad + 2\left(\frac{1}{r} + \frac{n-1}{R}\right)\phi_\ell(x_0) - \frac{1}{r}\phi_{\ell+1}(x_0) - \frac{1}{R}\phi_{\ell+1}(x_L) - \frac{n-2}{R}\phi_{\ell+1}(x') = 0, \\
-\frac{1}{r}\phi_{\ell-1}(x_L) - \frac{1}{R}\phi_{\ell-1}(x_0) - \frac{n-2}{R}\phi_{\ell-1}(x') \\
\quad + 2\left(\frac{1}{r} + \frac{n-1}{R}\right)\phi_\ell(x_L) - \frac{1}{r}\phi_{\ell+1}(x_L) - \frac{1}{R}\phi_{\ell+1}(x_0) - \frac{n-2}{R}\phi_{\ell+1}(x') = 0, \\
-\frac{1}{R}(\phi_{\ell-1}(x_0) + \phi_{\ell-1}(x_L)) - \left(\frac{1}{r} + \frac{n-3}{R}\right)\phi_{\ell-1}(x') \\
\quad + 2\left(\frac{1}{r} + \frac{n-1}{R}\right)\phi_\ell(x') - \frac{1}{R}(\phi_{\ell+1}(x_0) + \phi_{\ell+1}(x_L)) - \left(\frac{1}{r} + \frac{n-3}{R}\right)\phi_{\ell+1}(x') = 0.
\end{cases}
\tag{27}
$$

Finally, the last set (corresponding to $B^T\phi_{L-1} + D\phi_L = i_L$):

$$
\begin{cases}
-\frac{1}{r}\phi_{L-1}(x_0) - \frac{1}{R}\phi_{L-1}(x_L) - \frac{n-2}{R}\phi_{L-1}(x') + \left(\frac{1}{r} + \frac{n-1}{R}\right)\phi_{L-1}(x_0) = 0, \\
-\frac{1}{r}\phi_{L-1}(x_L) - \frac{1}{R}\phi_{L-1}(x_0) - \frac{n-2}{R}\phi_{L-1}(x') + \left(\frac{1}{r} + \frac{n-1}{R}\right)\phi_{L-1}(x_L) = -1, \\
-\left(\frac{1}{r} + \frac{n-3}{R}\right)\phi_{L-1}(x') - \frac{1}{R}(\phi_{L-1}(x_0) + \phi_{L-1}(x_L)) + \left(\frac{1}{r} + \frac{n-1}{R}\right)\phi_L(x') = 0.
\end{cases}
\tag{28}
$$

These equations can be significantly simplified by noting that $n = |\mathcal{X}| = S^D \gg 1$. In this case, $1/n \to 0$, and $n \approx n-1 \approx n-2 \approx n-3$. Then, multiplying the equations by $R/n$ and introducing $\widetilde{r} = r \cdot n/R$, we obtain:

$$
\begin{cases}
\left(\frac{1}{\widetilde{r}} + 1\right)\phi_0(x_0) - \frac{1}{\widetilde{r}}\phi_1(x_0) - \phi_1(x') = \frac{r}{\widetilde{r}}, \\
\phi_0(x') = \phi_1(x'), \\
\left(\frac{1}{\widetilde{r}} + 1\right)\phi_0(x_L) - \frac{1}{\widetilde{r}}\phi_1(x_L) - \phi_1(x') = 0.
\end{cases}
\tag{29}
$$

$$
\begin{cases}
\frac{1}{\widetilde{r}}\phi_{\ell-1}(x_0) - \phi_{\ell-1}(x') + 2\left(\frac{1}{\widetilde{r}} + 1\right)\phi_\ell(x_0) - \frac{1}{\widetilde{r}}\phi_{\ell+1}(x_0) - \phi_{\ell+1}(x') = 0, \\
\frac{1}{\widetilde{r}}\phi_{\ell-1}(x_L) - \phi_{\ell-1}(x') + 2\left(\frac{1}{\widetilde{r}} + 1\right)\phi_\ell(x_L) - \frac{1}{\widetilde{r}}\phi_{\ell+1}(x_L) - \phi_{\ell+1}(x') = 0, \\
-\phi_{\ell-1}(x') + 2\phi_\ell(x') - \phi_{\ell+1}(x') = 0.
\end{cases}
\tag{30}
$$

$$\begin{cases} -\dfrac{1}{\widetilde{r}}\phi_{L-1}(x_0) - \phi_{L-1}(x') + \left(\dfrac{1}{\widetilde{r}} + 1\right)\phi_L(x_0) = 0, \\[2mm] -\dfrac{1}{\widetilde{r}}\phi_{L-1}(x_L) - \phi_{L-1}(x') + \left(\dfrac{1}{\widetilde{r}} + 1\right)\phi_L(x_L) = -1, \\[2mm] \phi_{L-1}(x') = \phi_L(x'). \end{cases} \tag{31}$$

Hence:

$$\phi_0(x') = \phi_1(x') = \cdots = \phi_L(x').$$

Recall that the potential is defined up to an additive constant. Therefore, we are free to choose the reference point from which it is measured. Let us set:

$$\phi_\ell(x') = 0 \quad \forall \ell = 0, 1, \ldots, L.$$

Consequently, the system of equations simplifies further, decoupling into two independent parts:

System for $x_0$:

$$\begin{cases} \gamma\phi_0(x_0) - \phi_1(x_0) = r, \\ -\phi_{\ell-1}(x_0) + 2\gamma\phi_\ell(x_0) - \phi_{\ell+1}(x_0) = 0, \quad \text{for } \ell = 1, \ldots, L-1, \\ -\phi_{L-1}(x_0) + \gamma\phi_L(x_0) = 0. \end{cases} \tag{32}$$

System for $x_L$:

$$\begin{cases} \gamma\phi_0(x_L) - \phi_1(x_L) = 0, \\ -\phi_{\ell-1}(x_L) + 2\gamma\phi_\ell(x_L) - \phi_{\ell+1}(x_L) = 0, \quad \text{for } \ell = 1, \ldots, L-1, \\ -\phi_{L-1}(x_L) + \gamma\phi_L(x_L) = -r. \end{cases} \tag{33}$$

Here, $\gamma = 1 + \widetilde{r} = 1 + \frac{r \cdot n}{R}$. From the solution of (32), the solution of (33) can be obtained easily: $\phi_\ell(x_L) = -\phi_{L-\ell}(x_0)$.

Suppose that $\phi_\ell(x_L)$ is proportional to $\phi_0(x_L)$ via some polynomial of $\gamma$:

$$\phi_\ell(x_L) = P_\ell(\gamma)\phi_0(x_L). \tag{34}$$

Substituting this into (33), we find that the polynomial $P_\ell(\gamma)$ must satisfy the conditions:

$$\begin{aligned} P_0(\gamma) &= 1, \\ P_1(\gamma) &= \gamma, \\ P_{\ell+1}(\gamma) &= 2\gamma P_\ell(\gamma) - P_{\ell-1}(\gamma). \end{aligned} \tag{35}$$

These are the recurrence relations for the Chebyshev polynomials of the first kind:

$$P_\ell(\gamma) \equiv T_\ell(\gamma).$$

Substituting (34) into the last equation of (33), we obtain:

$$-\phi_{L-1}(x_L) + \gamma\phi_L(x_L) = \left(-T_{L-1}(\gamma) + \gamma T_L(\gamma)\right)\phi_0(x_L) = -r.$$

Thus:

$$\phi_0(x_L) = -\frac{r}{-T_{L-1}(\gamma) + \gamma T_L(\gamma)}$$

and the solutions are:

$$\phi_\ell(x_L) = -\frac{T_\ell(\gamma)\, r}{-T_{L-1}(\gamma) + \gamma T_L(\gamma)}, \quad \phi_\ell(x_0) = +\frac{T_{L-\ell}(\gamma)\, r}{-T_{L-1}(\gamma) + \gamma T_L(\gamma)}. \tag{36}$$

### A.2.5 CASE $x_0 = x_L$

In this case, similarly, $\phi_\ell(x') = 0$ for all $\ell = 0, 1, \ldots, L$, and the system of equations for $\phi_\ell(x_0) \equiv \phi_\ell(x_L)$ takes the form:

$$
\begin{cases}
\gamma\phi_0(x_0) - \phi_1(x_0) = r, \\
-\phi_{\ell-1}(x_0) + 2\gamma\phi_\ell(x_0) - \phi_{\ell+1}(x_0) = 0, \\
-\phi_{L-1}(x_0) + \gamma\phi_L(x_0) = -r.
\end{cases}
\tag{37}
$$

Suppose the solution has the form:

$$
\phi_\ell(x_0) = P_\ell(\gamma)\phi_0(x_0) - rF_\ell(\gamma).
\tag{38}
$$

Then the polynomials satisfy the recurrence relations:

$$
\begin{aligned}
P_{\ell+1}(\gamma) = 2\gamma P_\ell(\gamma) - P_{\ell-1}(\gamma), \\
F_{\ell+1}(\gamma) = 2\gamma F_\ell(\gamma) - F_{\ell-1}(\gamma)
\end{aligned}
\tag{39}
$$

with the initial conditions:

$$
P_0(\gamma) = 1, P_1(\gamma) = \gamma, \qquad F_0(\gamma) = 0, F_1(\gamma) = 1, F_2(\gamma) = 2\gamma.
\tag{40}
$$

This allows us to identify $P_\ell(\gamma)$ and $F_\ell(\gamma)$ with the Chebyshev polynomials of the first and second kind, respectively:

$$
P_\ell(\gamma) = T_\ell(\gamma), \quad F_\ell(\gamma) = U_{\ell-1}(\gamma).
$$

Note that the polynomial $U_\ell(\gamma)$ incorporates an additional condition $U_{-1}(\gamma) \equiv 0$, which is not usually implied in the standard definition.

Substituting (38) into the last equation of (37), we obtain:

$$
\phi_0(x_0) = r\frac{\gamma U_{L-1}(\gamma) - U_{L-2}(\gamma) - 1}{\gamma T_L(\gamma) - T_{L-1}(\gamma)}.
$$

Therefore, the final expression for the potentials is:

$$
\phi_\ell(x_0) = r\left(T_\ell(\gamma)\frac{\gamma U_{L-1}(\gamma) - U_{L-2}(\gamma) - 1}{\gamma T_L(\gamma) - T_{L-1}(\gamma)} - U_{\ell-1}(\gamma)\right).
\tag{41}
$$

This completes the proof of the Lemma.

## B EXPERIMENTAL DETAILS

In the 1D illustrative example §5.1, we use a simple MLP model with hidden layers of size [128, 128, 128] and ReLU activations. To condition on the layer number, we use a lookup table, specifically an embedding layer of size 2. We train the network with a learning rate of $2e - 4$, weight decay of $1e - 4$ during 5 000 training steps with batch size equals 256, and the SGD optimizer without a learning rate schedule. The network for the experiment in §5.2 shares the same architectures and optimization settings. Training the 1D and 2D experiment requires 2 minutes on a single A100 GPU (30 GB VRAM). The sampling time in 2D experiment for generation data with batch size 256 is 0.13 sec.

## C ABLATION STUDY

This appendix provides an empirical analysis of our method's key governing hyper parameter: the resistance ratio $\gamma = \frac{r}{R}$. This ratio is not merely a numerical hyperparameter but stems directly from the physical analogy underpinning our framework, where it controls the effective conductance of the optimization pathway.

The parameter $\gamma$ acts as a convergence rate modulator. We consider three different values of $\gamma$ and plot $\phi_\ell(x_\ell)$ (see Table 1) dependence on layer $\ell$ (Fig. 8) and corresponding marginal distributions at each $\ell$-th layer of pathway(Fig. 9).

**High-Conductance Regime** ($\gamma \rightarrow 1$, Fig. 8a): The system approaches a critically damped state, facilitating the fastest flow and yielding swift convergence in initial iterations (Fig. 9, the top image).

**Low-Conductance Regime** ($\gamma \rightarrow 0$, Fig. 8c): The process results in a slower flow compared to the previous example. Consequently, the method requires more iterations to converge to the target distribution (Fig. 9, the bottom image).

If we set $\gamma = 0.01$, we get the intermediate behavior of our model (Fig. 9, the middle image ). Thus, this relation $\gamma$ of resistances defines the rate of probability flow through the circuit.

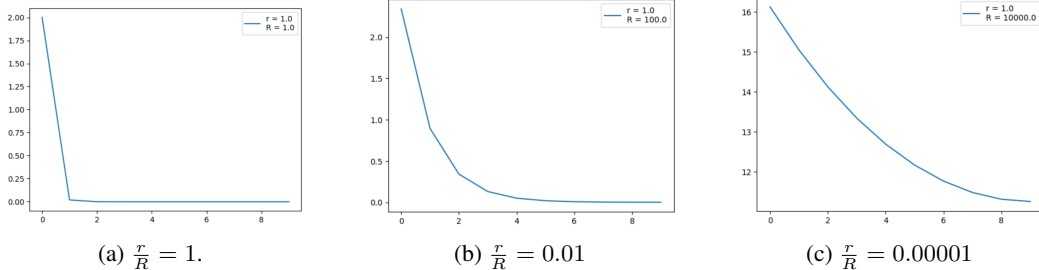

(a) $\frac{r}{R} = 1$.      (b) $\frac{r}{R} = 0.01$      (c) $\frac{r}{R} = 0.00001$

Figure 8: The convergence of potential $\phi_\ell(x_\ell)$ depends on the layer $\ell$ for three different values of $\gamma = \frac{r}{R}$ values.

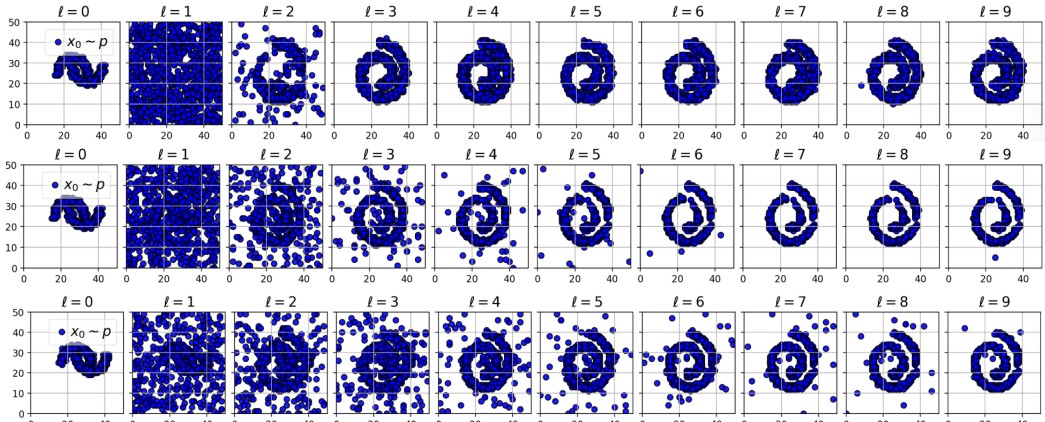

Figure 9: *Illustrative 2D Moons→Swiss Roll experiment*: input distribution $p(x_0)$ and the intermediate distributions at layers $\ell = \overline{1,9}$ and samples transformed from $x \sim p$ by $T(x)$ in a circuit $\mathcal{G}_{10}(2500, 1, 1)$ (**top** image), $\mathcal{G}_{10}(2500, 100, 1)$ (**middle** image) and $\mathcal{G}_{10}(2500, 10000, 1)$ (**bottom** image).

