# OpenReview forum: "Electric Currents for Discrete Data Generation"
_ICLR.cc/2026/Conference — Submitted to ICLR 2026_

### Official Review · Reviewer_MXYL · 2025-10-26

**Soundness:** 2
**Presentation:** 2
**Contribution:** 2
**Rating:** 4
**Confidence:** 2

**Summary:**

This paper mainly aims to answer the research question: How can we design a principled, physics-inspired method to transfer probability mass between two discrete data distributions and thereby generate discrete data, with guarantees that the transfer exactly matches the target distribution?

**Strengths:**

1. **[Important] ECD2G framework (electric-current analogy for discrete generation).** The paper models a discrete generative task as current flow on an L-partite graph, whose input layer carries the source distribution and output layer carries the target distribution. Kirchhoff’s current law and Ohm’s law connect node potentials and edge currents.
2. **[Important] Conceptually appealing design.** The conceptual move of mapping discrete distribution transfer to unit electrical flow with potentials/currents is neat and yields closed-form expressions for a particular circuit family. The transport-plan independence is conceptually appealing.

**Weaknesses:**

1. **[Important] Empirical validation seems to be purely qualitative and very small-scale.** Experiments are 1D U→N and 2D Moons→Swiss Roll grids; there appear to be no quantitative metrics (e.g., KL divergence), no ablations on R/r/L, and no comparisons to DDM/DFM/GFlowNets.

2. **[Important] Scalability concerns in high-dimensional discrete spaces.** The movement probabilities require summing currents over all outgoing edges with a dense next layer; this can be prohibitive. The authors also acknowledge the need for factorisation.

3. **Limited guidance on hyperparameter selection and sensitivity** (e.g., L/R/r). Qualitative comments exist (γ controls change rate), but there seems to be no systematic study.

**Questions:**

Refer to "Weaknesses"

---

> ### Author Response · Authors · 2025-12-03
> **Answer to the questions**
>
> Dear reviewer, we are grateful for your response and would like to comment on your questions further.
>
> **(Q1) [...]there appear to be no quantitative metrics (e.g., KL divergence), no ablations on R/r/L, and no comparisons to DDM/DFM/GFlowNets**
>
> **Quantitative comparisons.** It is worth noting that our primary goal in the experimental section is to demonstrate a **proof-of-concept** of our method. However, we agree that providing additional comparisons with other approaches enhances the understanding of its performance.
>
> Following your request, we  include a comparison of our approach (ECD$^{2}$G) with other methods [1, 2] and evaluate their performance on the *Swiss roll* dataset using the MMD metric [3] with 2048 samples over 5 runs. The results indicate that our method performs comparably to these baselines (see lines 415-425 of the **revised** paper). Please, take a look.
>
> **Ablation study.** We agree that an ablation study of our method would enhance the understanding of our methodology. We have addressed your question in a newly added **appendix C**, where we describe the influence of the hyperparameters on performance. Please, take a look.
>
> **(Q2) [...]The movement probabilities require summing currents over all outgoing edges with a dense next layer; this can be prohibitive. The authors also acknowledge the need for factorisation.**
>
>  It is worth noting that a core strength of our method's current formulation specifically, its integrated factorization scheme is already demonstrably effective, as shown in our proof-of-concept 2D experiments.
>
>  We fully agree with the reviewer that an exhaustive summation over all possible states would become computationally prohibitive in high-dimensional spaces. This critical challenge of developing an efficient sampling strategy that leverages the factorization scheme for scalability in such domains is a central focus of our planned future research. We view it as a highly promising avenue to extend the practical utility of the method's underlying principles in the future research.
>
>
>
>
>
> **Concluding remarks**
> We would be grateful if you could let us know if the explanations we gave have been satisfactory in addressing your concerns about our work. If so, we kindly ask that you consider increasing your score. We are also open to discussing any other questions you may have.
>
> **References**
>
>
> [1] - Austin et al, 2021(NeurIPS) "Structured Denoising Diffusion Models in Discrete
> State-Spaces"
>
> [2] - Gat et al, 2024(NeurIPS) "Discrete Flow Matching"
>
> [3] - Gretton et al, 2012(JMLR) "A kernel two-sample test"

---

### Official Review · Reviewer_FG64 · 2025-10-31

**Soundness:** 2
**Presentation:** 3
**Contribution:** 3
**Rating:** 2
**Confidence:** 3

**Summary:**

The paper introduces ECD2G, a novel generative modeling approach for discrete data that draws analogies between electric current flow in circuits and probability mass transport between distributions. It provides theoretical guarantees to prove that such transport 1) terminates in finite time and 2) correctly transports from the source distribution to the target distribution.While the theoretical framework is creative, the work suffers from significant practical limitations and insufficient experimental validation. It is only validated on toy datasets.

**Strengths:**

1. Novel theoretical framework: The perspective on combining electrical engineering principles into generative modeling is quite innovative and well-grounded in physics
2. Mathematical guarantees provided: Formal proofs for finite-step termination (Proposition 3.2) and correct transport (Corollary 3.3)
3. Conceptually clear framework: The L-partite graph structure and resistance parameterization are intuitive
4. Transport plan independence (Remark 3.4): Potential depends only on marginals, not on the coupling

**Weaknesses:**

1. Experimental validation is very limited: Model is only tested on trivial problems (1D: 50 states, 2D: 2,500 states) with no real-world applications despite claims about text, molecules, and images in the introduction
2. No quantitative metrics and no baseline comparisons: Only visual inspection of results, no distributional distance metrics, and zero comparison to discrete diffusion, flow matching, or other generative models
3. Fundamental scalability issue: Inference is bottlenecked by the denominator in Equation 9, which requires summing over S^D states—computationally infeasible for realistic problems
4. Factorization problem acknowledged but unsolved: Authors admit in Section 6 that scaling requires factorization, but provide no solution
5. Arbitrary design choices without justification: No guidance on selecting L, R, r parameters or circuit architecture

**Questions:**

1. Can the model be validated on real datasets such as ImageNet/CIFAR/MNIST or text generation benchmarks?
2. What is the computational cost of inference and training as a function of S, D, and L? How does it scale?
3. While innovative, what is the concrete superiority of this method compared to existing approaches like discrete diffusion models? What advantages justify the added complexity?
4. Why stop at D=2? Can you show results for D=3,4,5 to demonstrate where the method breaks down?
5. Can you provide even a preliminary factorization scheme to make the approach tractable beyond toy problems?

---

> ### Author Response · Authors · 2025-12-03
> **Answer to the questions**
>
> Dear reviewer, we sincerely appreciate your time and insightful review. We are delighted that you highlighted the innovative aspects of our approach and its independence from a transport plan. Our point-by-point responses follow.
>
> **(Q1) [...]what is the concrete superiority of this method compared to existing approaches like discrete diffusion models? What advantages justify the added complexity?**
>
> As described in Section 4 (see lines 335-340), diffusion models for discrete data types use a **predefined** transition matrix for the forward process and are suitable only for a **noise-to-data** scenario. In contrast, the forward process in our approach is only defined by the **data itself** and is suitable for **both** scenarios: data-to-data and noise-to-data. Below, we sum up differences our approach from diffusion models in the table.
>
>
>  |                          | Predefined process | Data-to-Data scenario |
> | :----------------------- | :----------------- | :-------------------- |
> | Diffusion Discrete Models| Yes                | No                    |
> | ECD$^2$G                 | No                 | Yes                   |
>
> **(Q2) No quantitative metrics and no baseline comparisons[...]**
>
> It is worth noting that our primary goal in the experimental section is to demonstrate a **proof-of-concept** of our method. However, we agree that providing additional comparisons with other approaches enhances the understanding of its performance.
>
> Following your request, we  include a comparison of our approach (ECD$^{2}$G) with other methods [1, 2] and evaluate their performance on the *Swiss roll* dataset using the MMD metric [3] with 2048 samples over 5 runs. The results indicate that our method performs comparably to these baselines (see lines 415-425 of the **revised** paper). Please, take a look.
>
>
> **(Q3) Arbitrary design choices without justification: No guidance on selecting L, R, r parameters or circuit architecture.**
>
> We agree that an ablation study of our method would enhance the understanding of our methodology. We have addressed your question in a newly added **appendix C**, where we describe the influence of the hyperparameters on performance.
>
> **(Q4) Why stop at D=2?[...]Experimental validation is very limited[...]Can the model be validated on real datasets such as ImageNet/CIFAR/MNIST[...]**
>
> As previously mentioned, our primary goal in the experimental section is to demonstrate a **proof-of-concept** of our method. Testing the method on more intricate and applied tasks would undoubtedly provide deeper insight into its capabilities and limitations.
> Further we believe that application of the method to more practical problems remains a promising avenue for future research.
>
>
>
>
> **(Q5)Fundamental scalability issue: Inference is bottlenecked by the denominator in Equation 9[...]Factorization problem acknowledged but unsolved[...]Can you provide even a preliminary factorization scheme to make the approach tractable beyond toy problems?[...] How does it scale?**
>
> It is worth noting that a core strength of our method's current formulation specifically, its integrated factorization scheme is already demonstrably effective, as shown in our proof-of-concept 2D experiments.
>
>  We fully agree with the reviewer that an exhaustive summation over all possible states would become computationally prohibitive in high-dimensional spaces. This critical challenge of developing an efficient sampling strategy that leverages the factorization scheme for scalability in such domains is a central focus of our planned future research. We view it as a highly promising avenue to extend the practical utility of the method's underlying principles in the future research.
>
> **Concluding remarks**.
> Thank you for your review. We hope our responses have satisfactorily addressed your concerns. If you feel your concerns have been resolved, we would be very grateful if you could consider raising your score. We are, of course, happy to discuss any further questions you may have.
>
>
> **References**
>
>
> [1] - Austin et al, 2021(NeurIPS) "Structured Denoising Diffusion Models in Discrete
> State-Spaces"
>
> [2] - Gat et al, 2024(NeurIPS) "Discrete Flow Matching"
>
> [3] - Gretton et al, 2012(JMLR) "A kernel two-sample test"

---

### Official Review · Reviewer_KU1o · 2025-10-31

**Soundness:** 2
**Presentation:** 2
**Contribution:** 2
**Rating:** 2
**Confidence:** 2

**Summary:**

The paper introduces ECD2G, a new generative model for discrete data inspired by electric current flow in circuits.

**Strengths:**

- The electric current-based generative models are novel and have the potential to be a good solution for discrete data generation.
- The overall methodology is explained clearly with a solid theoretical framework.

**Weaknesses:**

Generally I believe this work is promising but far from ready to publish yet, see the following comments:
- The motivation for using electric circuits to model the generative process seems somewhat arbitrary. It will be good if the authors could elaborate more on how they came to this idea.
- Some necessary comparison with other methods for discrete data generation is missing. The authors should either compare with the methods mentioned in lines 335-345, or explain why they are not comparable.
- Study on the efficiency compared to other methods is missing.
- Currently the experiments are based on very simple tasks. A study on some more practical tasks may be necessary.

**Questions:**

- Could the authors elaborate more on how they came to this idea (i.e. connecting data generation with electric circuits)?

---

> ### Author Response · Authors · 2025-12-03
> **Answer to the questions**
>
> Dear reviewer, thank you for taking the time to provide such a thoughtful review. We are very pleased that you see the potential of our method to be an effective solution for discrete data generation. In the following, we address your  questions.
>
> **(Q1) Some necessary comparison with other methods for discrete data generation is missing [...] Study on the efficiency compared to other methods is missing.**
>
> It is worth noting that our primary goal in the experimental section is to demonstrate a **proof-of-concept** of our method. However, we agree that providing additional comparisons with other approaches enhances the understanding of its performance.
>
>
> Following your request, we  include a comparison of our approach (ECD$^{2}$G) with other methods [1, 2] and evaluate their performance on the *Swiss roll* dataset using the MMD metric [3] with 2048 samples over 5 runs. The results indicate that our method performs comparably to these baselines (see lines 415-425 of the **revised** paper). Please, take a look.
>
>
>
> **(Q2)Currently the experiments are based on very simple tasks. A study on some more practical tasks may be necessary.**
>
>
>
> As previously mentioned, our primary goal in the experimental section is to demonstrate a **proof-of-concept** of our method. Testing the method on more intricate and applied tasks would undoubtedly provide deeper insight into its capabilities and limitations.
> Further we believe that application of the method to more practical problems remains a promising avenue for future research.
>
>
> **(Q3) The motivation for using electric circuits to model the generative process seems somewhat arbitrary[...]Could the authors elaborate more on how they came to this idea?**
>
> Thank you for this question. If we understand correctly, you are interested in the intuitive motivation for our approach.
>
> We would like to highlight the discussion in lines 63-70 of the paper, where we outline the fundamental gap that our method solves. Please, take a look.
>
>
>
> **Concluding remarks** We thank the reviewer for the raised concerns and questions. We hope that we have addressed them. If so, we kindly ask the reviewer to consider rising the score and confidence. We are ready to continue the discussion of the existing questions, and we are also happy to answer new ones.
>
>
> **References**
>
> [1] - Austin et al, 2021(NeurIPS) "Structured Denoising Diffusion Models in Discrete
> State-Spaces"
>
> [2] - Gat et al, 2024(NeurIPS) "Discrete Flow Matching"
>
> [3] - Gretton et al, 2012(JMLR) "A kernel two-sample test"

---

### Official Review · Reviewer_biis · 2025-11-01

**Soundness:** 2
**Presentation:** 2
**Contribution:** 2
**Rating:** 4
**Confidence:** 4

**Summary:**

The paper introduces Electric Current Discrete Data Generation (ECD$^2$G), a novel framework for generative modeling in discrete domains inspired by electrical engineering. The key idea is to model the transfer of probability mass between discrete data distributions as an electric current flow in a circuit. Input nodes represent samples from the source distribution, output nodes correspond to the target distribution, and the currents, learned via a neural network, encode the probability flow through the circuit. The authors develop a theoretical formulation grounded in Kirchhoff’s and Ohm’s laws, define a stochastic movement rule to realize the probability transfer, and propose a training algorithm that approximates the electric currents using regression.

**Strengths:**

1. The paper provides an extensive theoretical construction, linking discrete generation to well-established physical laws
2. The analogy with Kirchhoff’s current law ensures that probability mass is preserved during generation, offering a clear interpretability of the model’s constraints
3. Simple but consistent toy experiments show that the approach can approximately recover target distributions, supporting the conceptual feasibility

**Weaknesses:**

1. The mathematical notation is often ambiguous, making it difficult to follow key intuitions and derivations
2. While the physical analogy is interesting, the paper fails to convey why this analogy leads to practical or computational advantages over existing methods (e.g., discrete diffusion or flow matching)
3. Theoretical sections are disproportionately long and detailed, while experiments are extremely limited and do not convincingly demonstrate the model’s utility

**Questions:**

1. What are the computational or representational benefits of the electric circuit formulation compared to other discrete generative frameworks like Discrete Flow Matching or Discrete Diffusion Models?

2. A vertex $x_0 ∈X_{(0)}$ is defined as a (single) source vertex in $V^{(src)}$. Similarly, $x_L ∈X_{(L)}$ is a (single) sink in $V^{(snk)}$. What's the interpretation of $R_{\ell,\ell+1}(x_\ell,x_{\ell+1})$? I understood that $\ell$ is an index for layers, hence is $x_\ell$ a single node or all the nodes in a layer? In equation 11, what do you mean for $x_\ell = x_{\ell+1}$?

3. In equation 12, did you interpret the potential $\phi$ as a probability distribution? Are you exploiting the law of total probability? In that case, I would expected some integrals, since potentials are continuous. Which is the rationale of the equation?

4. The proposed electrical interpretation of optimal transport leaves me somewhat unconvinced for two main reasons:
- Defining, in probabilistic terms, the connection of a node $x$ with its nodes in $fan_{out}$ implies that its mass (or current intensity) would be transferred to only one of the output nodes. However, the physics of electrical circuits aligns more naturally with a fuzzy logic interpretation, since in reality the current would split and flow toward all output nodes.
- The statement \textit{``Remark 3.4 (Transport plan independence). Potential $\phi_\ell(x_\ell)$ is independent from $\pi(x_0, x_L)$''} does not correspond well to the physical behavior of electrical circuits. I admit, however, that since it's not clear to me the meaning of $x_0$ and $x_L$, it's difficult to discuss about it.
Could you better explain your intuition?

5. In equation 14 the ground truth is provided by a Monte Carlo method. May you better explain how to build a suitable network to address optimal transportation problems? May you also explain which is the benefit of your model since in any case there is the need of another estimation approach?

---

> ### Author Response · Authors · 2025-12-03
> **Answer to the questions**
>
> Dear reviewer, we are grateful for your response and would like to comment on your questions further.
>
> **(Q1) What are the computational or representational benefits of the electric circuit formulation compared to other discrete generative frameworks like Discrete Flow Matching or Discrete Diffusion Models?**
>
> As described in Section 4 (see lines 335-340), diffusion models for discrete data types use a **predefined** transition matrix for the forward process and are suitable only for a **noise-to-data** scenario. In contrast, the forward process in our approach is only defined by the **data itself** and is suitable for **both** scenarios: data-to-data and noise-to-data. Below, we sum up differences our approach from diffusion models in the table.
>
>
>  |                          | Predefined process | Data-to-Data scenario |
> | :----------------------- | :----------------- | :-------------------- |
> | Diffusion Discrete Models| Yes                | No                    |
> | ECD$^2$G                 | No                 | Yes                   |
>
> **(Q2) In equation 11, what do you mean [...]**
>
> It means that discrete state is not change through the layer. Since we can jump from one state at $\ell$-th layer to another at  $\ell+1$-th layer, this formulation means that the state is not changed.
>
> **(Q3) What's the interpretation of[...]**
>
>  This value represents the resistance between corresponding states in neighboring layers.
>
> **(Q4)experiments are extremely limited and do not convincingly demonstrate the model’s utility**
>
>  As previously mentioned, our primary goal in the experimental section is to demonstrate a proof-of-concept of our method. Testing the method on more intricate and applied tasks would undoubtedly provide deeper insight into its capabilities and limitations. Further we believe that application of the method to more practical problems remains a promising avenue for future research.
>
> **(Q5)In equation 12, did you interpret the potential[...]**
>
> Dear reviewer, if we correctly understand you, your ask is about absence of integrals in our derivations.
>
> Yes, you are correct in your intuition. In the context of our network model, we are indeed treating the normalized inverse of the potential (or a function thereof) as a discrete probability distribution that dictates the likelihood of interaction or state transition between nodes (or "states") in adjacent layers. We are not interpreting the raw potential $\phi$ itself as a distribution, but rather constructing a distribution from it.
>
> **Concluding remarks** We thank the reviewer for the raised concerns and questions. We hope that we have addressed them. If so, we kindly ask the reviewer to consider rising the score and confidence. We are ready to continue the discussion of the existing questions, and we are also happy to answer new ones.

---

### Official Review · Reviewer_BrAq · 2025-11-03

**Soundness:** 2
**Presentation:** 2
**Contribution:** 2
**Rating:** 2
**Confidence:** 3

**Summary:**

The paper proposes Electric Current Discrete Data Generation (ECD2G), a novel generative modeling framework for discrete data inspired by electrical circuit theory. The authors draw an analogy between the flow of electric current and the transfer of probability mass between data distributions. Using Kirchhoff’s and Ohm’s laws, they formalize a system where probability flows from a source distribution $p4 to a target distribution $q$ through a multi-layered circuit graph. Neural networks approximate the local currents between layers, enabling sampling and generation of discrete data via learned stochastic transitions.

**Strengths:**

* The paper introduces an idea that is well grounded and studied in physics (Kirchhoff's and Ohm's law)
* The paper is easy to follow

**Weaknesses:**

The research question is unclear to me: in the examples provided, it is unclear to me why the samples from the input distribution should follow the electrical flow to become samples of the target distribution. In contrast to the 2022 NeurIPS paper, the authors do not specify which partial differential equation has to be obeyed and why.

Also, what is the advantage of using a multi-layer architecture? Why is it difficult to learn this function directly?

**Questions:**

1. For what type of discrete datasets is the electric current model a good modelling assumptions?

2. Why is the method not compared to more standard approaches of learning latent embeddings for probability distribution transformations?

---

> ### Author Response · Authors · 2025-12-03
> **Answer to the questions**
>
> Dear reviewer, thank you for taking the time to provide review. Below, we address your questions.
>
>
> **(Q1) Why is the method not compared to more standard approaches of learning latent embeddings for probability distribution transformations?**
>
>
> Dear reviewer, if we correctly understand your question, you ask about comparison with other generative models that works in noise-to-data scenario.
>
> It is worth noting that our primary goal in the experimental section is to demonstrate a **proof-of-concept** of our method. However, we agree that providing additional comparisons with other approaches enhances the understanding of its performance.
>
> Following your request, we  include a comparison of our approach (ECD$^{2}$G) with other methods [1, 2] and evaluate their performance on the *Swiss roll* dataset using the MMD metric [3] with 2048 samples over 5 runs. The results indicate that our method performs comparably to these baselines (see lines 415-425 of the **revised** paper). Please, take a look.
>
>
>
> **(Q2) For what type of discrete datasets is the electric current model a good modelling assumptions?**
>
> Our method does not impose strict a prior assumptions on the dataset itself. This is evident in our 2D **proof-of-concept** experiment *Moons $\to$ Swiss Roll*: the model successfully learns the underlying geometric shape of the Swiss Roll data distribution without additional constraints.
>
> Thus, we believe our model to be a strong candidate for generating datasets where discrete states are not merely isolated categories, but are connected through meaningful low-dimensional manifolds.
>
>
>
> **(Q3) Also, what is the advantage of using a multi-layer architecture? Why is it difficult to learn this function directly?**
>
> If we understand correctly, your question is about why we use a multi-layer neural network instead of a single linear layer (matrix), is that right?
>
> As it known, multi-layer architecture  is crucial  because it allows to learn *highly non-linear and complex transformations* that are necessary to map between discrete data distributions, such as from *Moons* to *Swiss Roll*. A single linear layer can only represent simple, affine transformations, which are insufficient to capture the intricate geometric shapes and topological constraints of these datasets.
>
>
>
> **(Q4) [...]it is unclear to me why the samples from the input distribution should follow the electrical flow to become samples of the target distribution. In contrast to the 2022 NeurIPS paper, the authors do not specify which partial differential equation has to be obeyed and why.**
>
> Dear reviewer, we address your point regarding the NeurIPS 2022 paper, could you please provide the paper title, author names, or a more specific reference? This will allow us to provide a more precise response in our rebuttal.
>
>
>
> **Concluding remarks** We thank the reviewer for the raised  questions. We hope that we have addressed them. If so, we kindly ask the reviewer to consider rising the score and confidence. We are ready to continue the discussion of the existing questions, and we are also happy to answer new ones.
>
> **References**
>
> [1] - Austin et al, 2021(NeurIPS) "Structured Denoising Diffusion Models in Discrete
> State-Spaces"
>
> [2] - Gat et al, 2024(NeurIPS) "Discrete Flow Matching"
>
> [3] - Gretton et al, 2012(JMLR) "A kernel two-sample test"

---

### Meta-Review · Area_Chair_fxRU · 2025-12-19

**Summary:**

Reviewers thought the paper's drawn analogy to electrical currents and connections to physics were interesting in principle. However, they all unanimously agree that as a machine learning paper, the connections to generative modeling are not expressed sufficiently. Different reviewers mention slightly different aspects, but they all have major concerns concerning the advantages of the framing, the research question at hand, the computational (dis-)advantages and the lack of drawing parallels to existing generative models. The AC agrees that while the paper is an interesting proof of concept, more in-depth work is required to raise the paper to a level expected for publication at ICLR

**Reviewer Concerns:**

Reviewers concerns are all similar, with a larger set of the concerns revolving around the scope, motivation, lack of related work discussion, and insufficient description of (dis-)advantages over other framings. Another, perhaps slightly lesser concern, is the limited evaluation. Neither of these two aspects was resolved in the rebuttals, which primarily provide short comments regarding the reviewers' questions.

**Reviewer Scores:**

Reviewers unanimously recommended rejection with their initial scores ranging from 2,2,2 to 4,4. Although there was no possibility for a discussion, the rebuttals are generally rather short and respond primarily to the questions. The big concerns seem to be left unresolved and some messages emphasize that this work is a proof of concept. It is unlikely that these responses, or any prospectively ensuing discussion, would have thus swayed a larger set of the reviewers to a positive recommendation for this version of the paper.

---

### Decision · Program_Chairs · 2026-01-26

Reject